# EHR2Path: Scalable Modeling of Longitudinal Health Trajectories with LLMs

## Abstract

Healthcare systems face significant challenges in managing and interpreting vast, heterogeneous patient data for personalized care. Existing approaches often focus on narrow use cases with a limited feature space, overlooking the complex, longitudinal interactions needed for a holistic understanding of patient health. In this work, we propose a novel approach to patient pathway modeling by transforming diverse electronic health record (EHR) data into a structured representation and designing a holistic pathway prediction model, EHR2Path, optimized to predict future health trajectories. Further, we introduce a novel summary mechanism that embeds long-term temporal context into topic-specific summary tokens, improving performance over text-only models, while being much more token-efficient. EHR2Path demonstrates strong performance in both next time-step prediction and longitudinal simulation, outperforming competitive baselines. It enables detailed simulations of patient trajectories, inherently targeting diverse evaluation tasks, such as forecasting vital signs, lab test results, or length-of-stay, opening a path towards predictive and personalized healthcare. We will release our code upon acceptance.

## 1 Introduction

Healthcare systems face increasing challenges in managing vast amounts of patient data while striving for personalized, efficient care (Rajkomar et al., 2018; Jensen et al., 2012). The data collected and stored in healthcare has reached a scale where human interpretation alone is insufficient, necessitating the support of intelligent computational systems (Topol, 2019). So far, while the progress in computer-aided diagnosis has enhanced diagnostic accuracy and efficiency (Khalifa & Albadawy, 2024), most predictive models compartmentalize patient information, failing to capture the complex interactions of health factors over time (Rajkomar et al., 2018; Miotto et al., 2018). Recent efforts in Electronic Health Record (EHR) modeling have integrated structured and unstructured data, including demographics, textual reports, and time-series events (Yang et al., 2023a; Jensen et al., 2012; McDermott et al., 2021; Pellegrini et al., 2023a; Lovón-Melgarejo et al., 2024; Kraljevic et al., 2024). However, these studies often use a narrow set of features, often rely on complex data curation, and evaluate on limited tasks, leaving the potential for holistically modeling patient pathways largely unexplored.

Patient pathways (Richter & Schlieter, 2019) capture health trajectories at scales ranging from long-term, spanning a patient's lifetime, tackling e.g. chronic disease progression, to short-term, focused on specific healthcare episodes like hospital stays. This work focuses on short-term hospital trajectories, as visualized in Figure 1. A comprehensive in-hospital patient pathway model that can simulate future patient trajectories, could anticipate deteriorating vital signs, forecast adverse events in order to preemptively react to them, and optimize care pathways based on projected patient needs, supporting proactive, personalized care.

Recent advancements in multimodal large language models (MLLMs), have demonstrated strong capabilities in processing complex, multi-modal data (Anil et al., 2023; Touvron et al., 2023; OpenAI, 2023). In medicine, LLMs have excelled in tasks such as medical exam solving (Singhal et al., 2023b; Toma et al., 2023), literature comprehension (Singhal et al., 2023a), and conversational diagnosis (Li et al., 2023; Toma et al., 2023; Pellegrini et al., 2023b). Unlike AI models tailored for specific data types, LLMs can unify diverse inputs by transforming them into compatible representations, whether as textual, numerical, or categorical sequences, preserving contextual relationships.

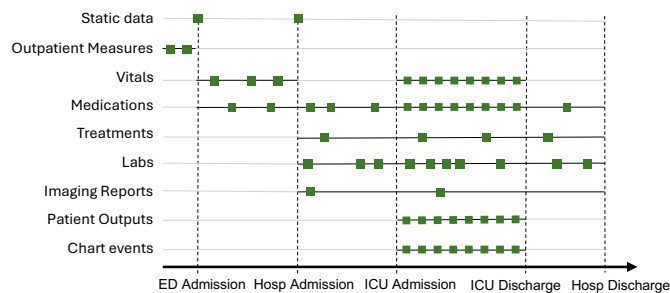

Figure 1: Short-term Patient Pathway visualization within Emergency Department, Hospital and ICU.

We believe that accurately modeling patient pathways demands models capable of integrating all available patient data over extended time periods. Towards this, we introduce EHR2Path, a novel patient pathway prediction model, extending over prior work in comprehensiveness and scale of considered large-scale EHR data, covering a patient's full stay in the hospital including emergency department, hospital, and intensive care unit (ICU) data. We transform a wide range of highly heterogeneous inputs into a unified, structured text representation that is effectively processed by building upon the knowledge and capabilities of LLMs. To capture highly detailed trajectories, we propose a novel summary mechanism that compresses past context into topic-specific "summary tokens," enabling up to 625× more context and a regularizing length-of-stay counter during inference. Finally, we ablate two complementary fine-tuning strategies to specialize EHR2Path to specific tasks. By demonstrating the strong performance of EHR2Path in patient trajectory forecasting and outcome prediction on the MIMIC-IV dataset (Johnson et al., 2023b), we establish a scalable methodology for comprehensive modeling of patient trajectories, advancing personalized and predictive healthcare.

## 2  RELATED WORK

The integration of heterogeneous EHR data into machine learning models has been a significant research focus, particularly with the adoption of sequence models like transformers (Vaswani, 2017). Existing methods fall into two categories: *Outcome Prediction Models*, that target specific patient outcomes, such as mortality, length-of-stay or diagnosis prediction, and *Patient Timeline Prediction Models*, which forecast full health trajectories but remain under-explored.

**Outcome Prediction Methods**  Most EHR models predict outcomes using structured data like medical codes. Models such as BEHRT (Li et al., 2020), Med-BERT (Rasmy et al., 2021), CEHR-BERT (Pang et al., 2021), TransformEHR (Yang et al., 2023b) have enhanced accuracy with temporal embeddings, artificial time tokens and multi-task learning. EHRSHOT (Wornow et al., 2023) introduced a benchmark integrating structured demographics and various coded events. Some other works include additional numerical or categorical data (Li et al., 2022; Pellegrini et al., 2023a; Lovón-Melgarejo et al., 2024). Another major challenge in EHR modeling is *Long-Context Integration*, due to EHR's time span and volume. Some outcome prediction models like Hi-BEHRT (Li et al., 2022), EhrMamba (Fallahpour et al., 2024), and CONTEXT (Wornow et al., 2024) extend token limits with hierarchical or sub-quadratic architectures, while models like REMed (Kim et al., 2023) and EMERGE (Zhu et al., 2024) propose retrieval-augmented approaches for selecting relevant features. UniHPF and GenHPF (Hur et al., 2022; 2023) show the effectiveness of combining structured and unstructured data into a unified text-based event representation with promising results on outcome classification tasks. Despite these advances, outcome prediction models typically target rather narrow downstream tasks and operate on a limited feature space, often focusing only on a fixed set of structured medical codes or few continuous signals, excluding a large set of available EHR data. They generally do not model or generate full patient trajectories.

**Patient Timeline Prediction Methods**  In contrast to the prevalent outcome prediction methods, patient timeline prediction seeks to model and predict the entire sequence of a patient's health events, offering a more holistic view of health trajectories. For instance, ESGPT (McDermott et al., 2023) provides a flexible library for transformer-based modeling of continuous-time event streams, including

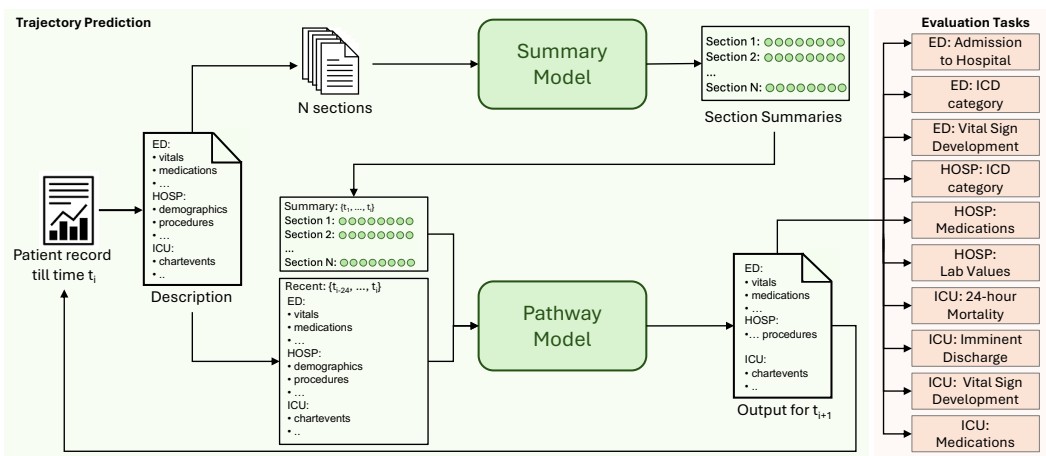

Figure 2: Overview of our proposed method. A patient record is structured into text from which a fixed time window is kept as text representation, while the full temporal context is summarized into an embedding-based summary by the summary model. The Pathway Model combines both representations to predict the next time-step. For iterative simulation, predictions update the patient record to simulate future trajectories until a termination condition is met.

pre-processing utilities and example architectures. Recent approaches like CEHR-GPT (Pang et al., 2024) adapt GPT models for synthetic EHR generation, using structured tokens enriched with lab results, temporal and demographic information, but do not target patient-specific clinical forecasting. MOTOR (Steinberg et al., 2023) trains on timestamped sequences of medical events for time-to-event prediction, targeting individual outcomes like death or lab tests. Foresight (Kraljevic et al., 2024) predicts biomedical concepts from clinical notes, converting text into structured sequences using the SNOMED ontology (El-Sappagh et al., 2018). ETHOS (Renc et al., 2024) models structured EHR data from MIMIC-IV, generating tokenized Patient Health Timelines (PHTs). Yet, it omits unstructured notes and ICU chart events, key sources of patient information, and is limited to a 2,048-token input window. While these methods represent important progress toward trajectory modeling, they often operate on constrained input lengths, exclude unstructured modalities such as free-text notes and dense time-series signals such as ICU chart events and do not assess long-term simulation capabilities.

Our work shifts the focus to forecasting and evaluating on entire hospital pathways, moving beyond isolated outcome prediction towards a comprehensive understanding of patient development. Unlike prior methods, we comprehensively integrate all available structured and unstructured EHR data and leverage LLMs to understand this heterogeneous and noisy information in its native form. To address long-context challenges, we introduce a novel summary mechanism that enables efficient modeling over extended time horizons and complex patient histories.

## 3 SCALABLE MODELING OF HEALTH TRAJECTORIES

We propose a novel approach to predict longitudinal patient health trajectories by integrating heterogeneous EHR data into a structured, language-based representation. Using an LLM backbone, we leverage its contextual understanding to model complex health data. To capture extended time horizons, we introduce a Masked Summarization Bottleneck that efficiently compresses longitudinal data. Additionally, we introduce two fine-tuning strategies for specializing EHR2Path to specific tasks. Our method enables comprehensive trajectory simulation, supporting holistic patient modeling.

### 3.1 DATA REPRESENTATION

EHR data is inherently heterogeneous, including static and temporal data, numerical and categorical features, and unstructured text from various clinical settings. We develop a unified textual representa-

tion of this data preserving its original form, avoiding heavy pre-processing to embrace real-world noise and incompleteness. This enhances robustness and supports diverse inputs without feature-specific adjustments, thereby improving scalability and real-world applicability. Our method supports diverse data types and uses natural language for feature names and values, instead of medical codes, enabling semantic interpretation by an LLM. An exemplary data sample is shown in Appendix D.

**Hierarchical Data Organization**   We denote the recorded data for patient $p$ as $D_{p,t} \mid t \in T_p$, where $t$ denotes the current hour of the stay and $T_p$ the total stay time of patient $p$. $D_{p,t}$ is hierarchically organized into three levels:

***Clinical Units***: At the highest level, $D_{p,t}$ is partitioned into clinical units $U = \{\text{ED}, \text{Hospital}, \text{ICU}\}$, representing the Emergency Department, Hospital, and Intensive Care Unit.

***Data Categories***: At each clinical unit $u \in U$, the data is subdivided into categories $C_u$ such as Demographics, Vital Signs, Prescriptions, Procedures, Radiology Reports, and Chart Events.

***Features***: Each category $c \in C_u$ consists of one or multiple features $F_{u,c} = \{f_1, f_2, \ldots, f_n\}$. We consider two types of features, categorized as *Events* or *Values*. *Events*, e.g. treatments or reports, do not record values and are directly included in the data representation. *Values* require value recording and may represent binary indicators *(e.g., "ST Segment Monitoring On: yes/no")*, continuous numerical values *(e.g., "Heart Rate: 82")* or categorical variables *(e.g., "Heart Rhythm: Sinus Rhythm")*, capturing the diversity of EHR data.

**Temporal Features**   Most information in a patient's EHR varies over time. Such temporal data is modeled as a sparse feature sequence $\mathbf{Z}_f = \{z_{f,t} \mid t \in \{1, 2, \ldots, T_p\}\}$, where each $z_{f,t}$ represents the recorded value or event at timestamp $t$, where $t$ specifies how many hours have passed since the value was recorded relative to the current time-point (e.g., *"Heart Rate: 1: 80"* for the previous hour). Consecutive identical values are merged into intervals (e.g., *"Heart Rate: 10–4: 82"*). Missing data is indicated by the absence of a corresponding entry in $\mathbf{Z}_f$ *(e.g., "Heart Rate: 5:82, 1:80")*.

**State Attributes and Diagnosis Codes**   We incorporate state attributes to mark significant changes in the patient's trajectory. These include events such as discharge from the ED, admission to and discharge from the ICU and Hospital, as well as death. At the final time step $T_p$ of a stay, ICD categories (Slee, 1978) corresponding to the patient's diagnosis are added.

**Input and Output Preparation**   Model input and output are constructed by generating textual representations following the hierarchical structure defined above. The input includes all available data for a time window between $t - w$ up to the current time-point $t$, where $w$ denotes the input window size. To integrate longer time windows, we introduce a *Masked Summarization Bottleneck*, which condenses the entire patient history into one or multiple embedded representations for each category consisting of $N$ summary tokens each, allowing a window size of $t$. The output is formatted similarly but exclusively contains data recorded at the subsequent time-point $t + 1$. To enhance efficiency, the output is sparse, including only the features that were actively recorded at $t + 1$.

## 3.2   Patient Trajectory Prediction

Building upon the structured, language-based EHR representation outlined in Section 3.1, we now detail our approach for predicting future patient states and clinical pathways.

### 3.2.1   Model Architecture and Training

The core component of our architecture is the *Pathway Model*, a transformer-based LLM tailored for patient trajectory prediction. It is trained to predict a sparse, structured textual output for the subsequent time-point $t + 1$, including all actively recorded features $z_{f,t+1}$ (e.g., lab values, procedures) and state transitions (e.g., discharge, transfer, or death), leveraging all available data from the patient record $D_{p,t}$. The training objective is to forecast all EHR elements recorded at $t + 1$, fostering a comprehensive understanding of the patient's clinical pathway. We propose and compare three model variants, each utilizing different inputs to the pathway model:

**EHR2Path-Text (E2P-T)**: Processes a text representation of static patient data and events within the most recent $w$ hours (e.g., $[t - w, t]$), capturing detailed recent observations in a structured format.

**EHR2Path-Summ (E2P-S)**: Relies on summary embeddings of the entire patient history, generated via the *Masked Summarization Bottleneck* (Section 3.2.2), efficiently integrating long-term context.

**EHR2Path-Summ+Text (E2P-S+T)**: Combines the text representation of recent events and the full

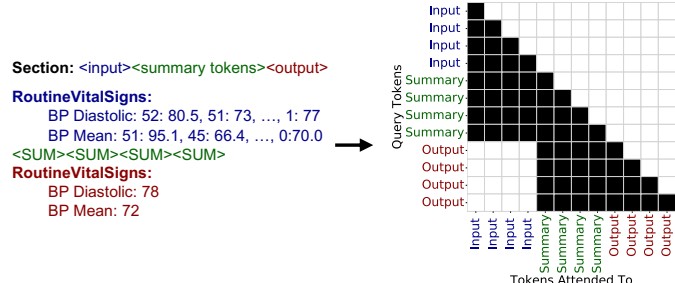

Figure 3: Masked Summarization Bottleneck. Input tokens encode past observations, while summary tokens (<SUM>) compress key information. A custom attention mask ensures outputs attend only to summaries, forcing the model to encode relevant patient data into a compact representation.

history summary, balancing detailed short-term data with comprehensive longitudinal context. Appendix E provides detailed implementation and training details.

**Inference for Trajectory Simulation** To simulate full patient trajectories, the Pathway Model is applied iteratively at each time step $t$ until a stopping criterion is met, such as hospital discharge, death, or a predefined number of steps. At each iteration, the Pathway Model predicts the structured output for $t + 1$, which is reintegrated into the patient record, yielding $D_{p,t+1}$. The new patient record $D_{p,t+1}$ is processed according to the model variants to form a new input. This iterative procedure generates a synthetic sequence of future states $\{z_{f,t+1}, z_{f,t+2}, \dots\}$ for all relevant features $f$, effectively simulating the patient's clinical course.

### 3.2.2 MASKED SUMMARIZATION BOTTLENECK

For *E2P-S* and *E2P-S+T*, we propose a novel *Masked Summarization Bottleneck* that produces compact, task-relevant representations for each clinical unit $u$ and data category $c$ to handle extensive patient histories. Formally, let $\mathbf{X}_t = \{x_1, x_2, \dots, x_n\}$ be the $n$ token long description of a section in the patient's record at time $t$. We append $m$ *summary tokens* $\mathbf{S} = \{s_1, \dots, s_m\}$ and $o$ *output tokens* $\mathbf{Y} = \{y_1, \dots, y_o\}$ describing the values at $t + 1$ for the current section, forming the sequence $\mathbf{X}_t^{\text{full}} = \{\mathbf{X}_t, \mathbf{S}, \mathbf{Y}\}$. A custom attention mask $\mathbf{M}$ constrains how tokens $i$ can attend to tokens $j$:

$$
M_{ij} = \begin{cases} 1 & \text{if } j \leq i \text{ and } i \leq n + m, \\ 1 & \text{if } j \leq i \text{ and } j > n, \\ 0 & \text{otherwise.} \end{cases}
$$

As visualized in Figure 3, this ensures that output tokens $\{y_1, \dots, y_o\}$ can only attend to the summary tokens $\{s_1, \dots, s_m\}$ and preceding output tokens, creating a bottleneck. Following the information bottleneck principle (Tishby et al., 1999), our approach derives a compact representation $\mathbf{S}$ that maximizes mutual information $I(\mathbf{S}; \mathbf{z}_{t+1})$ with the next state $\mathbf{z}_{t+1}$ while constraining $I(\mathbf{S}; \mathbf{Z}_{1:t})$ with the patient history $\mathbf{Z}_{1:t}$ subject to the capacity $m$. Consequently, the Summarization Module is driven to compress the most relevant information to predict future states into the summary tokens, while discarding uninformative or redundant details. During inference, the summary tokens provide a lightweight embedding of the entire patient history for each data category $c$ in $D_{p,t}$. By handling each section separately, this allows us to in most cases include the entire patient history. At inference, the input tokens $\{x_1, \dots, x_n\}$ for each category $c$ are combined with the summary tokens $\{s_1, \dots, s_m\}$, and a single forward pass computes their hidden states $\mathbf{H} = \{\mathbf{h}_1, \dots, \mathbf{h}_{n+m}\}$. The hidden states of the summary tokens, $\mathbf{H}^{\text{summary}} = \{\mathbf{h}_{n+1}, \dots, \mathbf{h}_{n+m}\}$ are the final output of the Summary Model. This enables efficient summarization of all sections in $D_{p,t}$ without separate models, thus reducing computational overhead. Unlike traditional auto-encoder approaches that reconstruct every detail of the input, we optimize summary tokens for forecasting the patient's next state. This task-focused approach avoids the inefficiencies associated with generic reconstructions and produces more effective representations for patient pathway prediction.

Table 1: Next Time-Step Prediction: We report event F1, event-value scores for numerical and categorical predictions and the avg./max. context and input tokens required to capture this context. 95% confidence intervals are given in brackets.

| | F1 macro/micro | Numerical ↓ macro/micro | Categorical macro/micro | Avg./Max. context | Avg./Max. input ↓ |
|---|---|---|---|---|---|
| Statistic | 0.02 / 0.02 | 0.99 / 0.97 | 0.03 / 0.16 | n/a | n/a |
| ETHOS | 0.003 (0.00,0.008) / 0.03 (0.03,0.03) | 1.00 (0.99,1.00) / 0.97 (0.97,0.97) | 0.04 (0.00,0.12) / 0.31 (0.31,0.32) | 1217 / 2048 | 1217 / 2048 |
| E2P-T-1h | 0.42 (0.30,0.53) / 0.64 (0.64,0.64) | 0.67 (0.52,0.80) / 0.89 (0.88,0.89) | 0.30 (0.16,0.44) / 0.52 (0.51,0.53) | 380 / 3142 | 380 / 3142 |
| E2P-T-24h | 0.47 (0.36,0.57) / **0.78** (0.78,0.78) | **0.64** (0.51,0.77) / 0.74 (0.74,0.75) | **0.33** (0.21,0.47) / **0.57** (0.56,0.57) | 1365 / 4250 | 1365 / 4250 |
| E2P-S | **0.48** (0.36,0.59) / 0.76 (0.76,0.77) | 0.66 (0.55,0.78) / **0.72** (0.71,0.72) | **0.33** (0.21,0.46) / 0.56 (0.55,0.56) | **9823 / 86621** | **220 / 1494** |
| E2P-T+S | 0.43 (0.33,0.54) / **0.78** (0.78,0.78) | 0.68 (0.58,0.79) / 0.73 (0.72,0.73) | **0.33** (0.21,0.46) / 0.55 (0.55,0.56) | **9823 / 86621** | **220 / 1494** |
| Restricted to data supported by ETHOS: | | | | | |
| ETHOS | 0.04 (0.00,0.08) / 0.08 (0.08,0.09) | 0.96 (0.96,0.96) / 0.96 (0.96,0.96) | 0.86 (0.86,0.86) / 0.85 (0.84,0.86) | 1217 / 2048 | 1217 / 2048 |
| E2P-T+S | **0.12** (0.00,0.24) / **0.24** (0.23,0.25) | **0.87** (0.87,0.87) / **0.87** (0.87,0.88) | **0.99** (0.99,0.99) / **0.93** (0.93,0.94) | **9823 / 86621** | **220 / 1494** |

### 3.2.3 LENGTH-OF-STAY INDICATOR

Converging to final states like discharge requires accurate state predictions across multiple forecast iterations, which is challenging because such events are relatively rare compared to typical value changes, often causing the model to inadvertently prolong patient stays without termination. We tackle this by appending a Length-of-Stay (LOS) indicator, a simple countdown of remaining hours for each clinical unit $u$. To increase robustness, and allow the model to work without a ground truth LOS Token in inference time, we perform two augmentations. In half of the samples, we drop the LOS token entirely from the input, and in the other half, we inject noise (+-20%) into the LOS token, encouraging the model to re-estimate the remaining LOS at each time step rather than rigidly decrementing it. Crucially, at inference time, we never include the ground truth LOS token in the input, instead the first step is prompted without LOS token, while in later steps the predicted LOS tokens of prior steps is included.

### 3.2.4 FINE-TUNING FOR OUTCOME PREDICTION

While our primary focus is on trajectory simulation, the model can also serve as a foundation for downstream prediction tasks through targeted fine-tuning. We explore two complementary fine-tuning strategies: *Specialized Pathway Fine-Tuning*, fine-tuning on curated pathway data in the original format (e.g., only Emergency Department cases), to specialize for specific outcomes while retaining insights into intermediate events; and *Outcome-Oriented Fine-Tuning*, simplifying the task to a single-step outcome prediction to maximize predictive performance.

## 4 EXPERIMENTAL SETUP

### 4.1 DATASET AND PRE-PROCESSING

We use the MIMIC-IV database (Johnson et al., 2023b), containing de-identified, real-world EHRs for approximately 300,000 patients, as it is one of the largest and the most comprehensive EHR database to date, offering granular and heterogeneous patient records, including coded, numerical and free-text features from different clinical units. We incorporate all clinically relevant tables, encompassing 22 tables and 20 ICU Chart Event categories (listed in Appendix A), including admissions, lab results, medications, procedures, vital signs, and more, spanning the Emergency Department (ED), hospital wards, and Intensive Care Unit (ICU). Further, its inherent noisiness and sparsity further strengthen its value as a realistic benchmark. To retain this real-world complexity, we avoid applying exclusion criteria and preserve missing or incorrectly populated fields. Following previous work (Wang et al., 2020; McDermott et al., 2021), we aggregate values hourly using average for numerical values,

and most frequent for categorical ones, managing computational feasibility by balancing temporal resolution and recording frequency. Average context lengths are 380 tokens per hour, 1,880 for 24 hours, or 11,361 for the full history. Data is split at the patient level: 95% for training, 2.5% each for validation and test sets. Training samples consist of EHR data $D_{p,t}$ up to time $t$, with $t+1$ as the label. In practice, since patient stays often span hundreds of hours, using all hourly slices is computationally infeasible. Instead, we sample time points from all patients using **weighted sampling**, where time points are weighted based on the rarity of clinical events in the output, with rarer events receiving higher log-scaled weights. Additionally, we **oversample** critical transitions and key events such as admissions, discharges, and deaths to ensure adequate representation in training. Overall, we collect one million weighted samples for training, while fixed sets of 5000 val/test time points each are sampled without weighting, preserving the original event distribution.

## 4.2 LONGITUDINAL SIMULATION TASKS

We define nine clinically relevant evaluation tasks across ED, Hospital, and ICU, focusing on outcome and development prediction, partially inspired by McDermott et al. (2021). For each task we perform 500 simulations to assess iterative trajectory prediction. ED tasks use only ED data, while Hospital and ICU tasks incorporate all prior data. Detailed task descriptions can be found in Appendix G.
**Outcome Prediction Tasks**: ED Admission *(will ED patient be hospitalized)*, ED/Hospital Discharge Diagnosis (multi-label prediction over 18 ICD categories), Imminent Mortality (24 hours), Imminent Discharge (3 days).
**Development Prediction Tasks**: 24-hour forecasts for ED Vital Signs, Hospital Medications, Hospital Lab Values, ICU Vital Signs, ICU Inputs.

## 4.3 BASELINES

For comparison, we select trajectory and outcome prediction methods trained on the MIMIC-IV dataset with publicly available code, enabling reproduction on our data splits and tasks: **ETHOS** (Renc et al., 2024), a transformer-based model for tokenized health timeline simulation, trained on next-token prediction. We compare against ETHOS on next time-step prediction and all applicable trajectory simulation tasks. **MEME** (Lee et al., 2024), an LLM-based approach for outcome classification on textualized Emergency Department data, fine-tuned to predict outcomes from the full textual summary of ED records. **REMed** (Kim et al., 2024), a transformer-based long context model for ICU outcome prediction that scores EHR events using an MLP and feeds the most important ones to a transformer for classification. We compare fine-tuned EHR2Path variants with MEME and REMed on applicable outcome prediction tasks.

## 4.4 EVALUATION METRICS

We assess performance on next-timestep prediction and longitudinal simulation using tailored metrics:

**Next-Timestep Prediction Metrics.**
*Event Prediction:* To accurately represent false positive and negative rates, we use micro- and macro-averaged F1 scores for predicting which events or features will be present in the next hour.
*Event+Value Prediction:* Measures overall correctness by requiring both correct event timing and value prediction. Numerical values use a modified MAE (correct-time predictions scored by MAE; incorrect/missing predictions receive maximum error). Numerical values are normalized with 1st–99th percentile clipping and min-max scaling to [0,1]. Categorical values use modified accuracy where missing predictions are counted as incorrect.
Macro metrics are averaged at feature level to provide an unbiased assessment across diverse features. Additionally we report the average and maximum captured context and required input tokens, measured as tokens in the tokenized sequence. Context refers to the token count of the text representation of the input window, and input tokens to the final model input after potential summarization.

**Longitudinal Simulation Metrics.** For extended trajectory forecasting, we use task-specific metrics. Binary classification tasks (e.g., mortality prediction) are evaluated using accuracy on balanced test sets, while regression-based tasks (e.g., vital sign forecasting) are assessed using event-based F1, MAE on min-max normalized values, and accuracy. For binary outcome tasks evaluated with

Table 2: Simulation-based Results.

|  | ETHOS | E2P-T | E2P-S | E2P-T+S |
|---|---|---|---|---|
| *ED Vital Sign Development* | | | | |
| Event F1 | n/a | **0.61** (0.58,0.64) | 0.53 (0.49,0.56) | 0.57 (0.54,0.60) |
| Value MAE ↓ | n/a | **0.10** (0.10,0.10) | 0.12 (0.12,0.13) | **0.10** (0.10,0.11) |
| Value Acc. | n/a | **0.54** (0.48,0.61) | **0.54** (0.46,0.61) | 0.51 (0.44,0.57) |
| *ED Admission Prediction* | | | | |
| Acc. | n/a | 0.67 (0.63,0.71) | 0.63 (0.59,0.67) | **0.68** (0.64,0.72) |
| *ED Discharge Diagnosis* | | | | |
| F1 | n/a | **0.41** (0.37,0.45) | 0.28 (0.24,0.32) | 0.38 (0.33,0.42) |
| *Hospital Medication Development* | | | | |
| Event F1 | n/a | 0.83 (0.80,0.85) | 0.82 (0.79,0.85) | **0.83** (0.80,0.85) |
| *Hospital Lab Value Development* | | | | |
| Event F1 | 0.05 (0.03,0.07) | 0.20 (0.16,0.25) | 0.17 (0.13,0.20) | **0.23** (0.19,0.27) |
| Value MAE ↓ | 0.21 (0.19,0.23) | 0.09 (0.08,0.09) | 0.11 (0.11,0.12) | **0.08** (0.08,0.09) |
| Value Acc. | n/a | 0.32 (0.00,1.00) | 0.36 (0.00,1.00) | **0.43** (0.13,0.81) |
| *Hospital Discharge Diagnosis* | | | | |
| F1 | n/a | 0.49 (0.47,0.51) | 0.47 (0.45,0.49) | **0.50** (0.48,0.52) |
| *ICU Vital Sign Development* | | | | |
| Event F1 | n/a | 0.70 (0.67,0.72) | **0.75** (0.73,0.78) | 0.71 (0.69,0.73) |
| Value MAE ↓ | n/a | 0.09 (0.09,0.09) | 0.10 (0.09,0.10) | **0.08** (0.08,0.08) |
| Value Acc. | n/a | 0.69 (0.66,0.72) | **0.80** (0.77,0.82) | 0.79 (0.77,0.81) |
| *ICU Input Development* | | | | |
| Event F1 | n/a | 0.79 (0.76,0.81) | **0.85** (0.83,0.87) | 0.84 (0.82,0.86) |
| *ICU Imminent Mortality* | | | | |
| Acc. | **0.61** (0.57,0.65) | 0.53 (0.49,0.58) | 0.50 (0.46,0.54) | 0.57 (0.53,0.62) |
| *ICU Imminent Discharge* | | | | |
| Acc. | *0.63* (0.51,0.74) | 0.63 (0.59,0.67) | 0.57 (0.53,0.61) | **0.69** (0.65,0.73) |

Table 3: Fine-Tuning for Outcome Tasks.

|  | MEME | REMed | E2P-FT-P | E2P-FT-O |
|---|---|---|---|---|
| *ED Admission Prediction* | | | | |
| Acc. | 0.67 (0.63,0.71) | n/a | 0.73 (0.69,0.77) | **0.74** (0.70,0.77) |
| *ED Discharge Diagnosis* | | | | |
| F1 | 0.42 (0.37,0.48) | n/a | (0.45)* (0.41,0.48) | **0.45** (0.41,0.48) |
| *Hospital Discharge Diagnosis* | | | | |
| F1 | n/a | n/a | (0.59)* (0.56,0.61) | 0.59 (0.56,0.61) |
| *ICU Imminent Mortality* | | | | |
| Acc. | n/a | 0.71 (0.60,0.80) | 0.77 (0.74,0.81) | **0.83** (0.80,0.86) |
| *ICU Imminent Discharge* | | | | |
| Acc. | n/a | **0.82** (0.78, 0.85) | 0.69 (0.65,0.73) | 0.76 (0.72,0.79) |

*FT-P: Specialized Pathway Fine-Tuning.*
*FT-O: Outcome-Oriented Fine-Tuning.*
*n/a: Indicates the model is not applicable to the task as it cannot integrate or predict the relevant data.*
*95% confidence intervals are given in brackets.*
*\*Diagnosis prediction is inherently a one-step task making pathway- and outcome-fine-tuning identical.*

accuracy, we use a balanced test set for sample efficiency. Value predictions are only evaluated if a matching time-point exists, with a $\pm 1$ hour tolerance to account for EHR recording variability.

## 5 RESULTS AND DISCUSSION

### 5.1 NEXT TIMESTEP PREDICTION RESULTS

We evaluated our models' ability to predict the next time-step with results in Table 1. We compare our models against a statistical baseline (sampling events by occurrence ratios and using mean/majority values) and ETHOS (Renc et al., 2024), a transformer-based timeline simulator. The statistical baseline's inferior results highlight task complexity. When evaluated on the full MIMIC data, ETHOS under-performs significantly across all metrics, mainly due to its inability to predict many EHR event types, which is reflected mainly in a very low F1 score. When restricting the evaluation to data supported by ETHOS, ETHOS improves slightly, but E2P-T+S still outperforms, even though it predicts a much wider range of data. Among our models, E2P-S achieves the highest macro F1 and

micro numerical score, demonstrating the effectiveness of our Masked Summarization Bottleneck in capturing essential information from the entire patient history. E2P-T-24h shows strong performance in micro F1, indicating its ability to handle frequent events and outperforms E2P-T-1h, showing the benefit of including more longitudinal context. The combined model, E2P-T+S, performs comparably to E2P-T-24h but does not surpass E2P-S in overall performance. Additionally, E2P-S and E2P-T+S handle significantly larger contexts, up to $20\times$ in MIMIC-IV. This shows our summarization mechanism effectively condenses extensive patient histories into manageable embeddings, improving prediction performance and enabling efficient handling of large-scale, heterogeneous EHR data.

## 5.2 PATIENT TRAJECTORY SIMULATION RESULTS

E2P-T-24h, E2P-S, and E2P-T+S all perform strongly on simulation tasks (Table 2), with distinct strengths: E2P-S excels in long-term, dense-context tasks such as ICU Vital Sign and Input Development, while E2P-T-24h outperforms in immediate state predictions such as ED Admission or Diagnosis tasks. The combined model, E2P-T+S consistently ranks first or second, often achieving the best results. This stability makes it a robust compromise, balancing text- and summary-based strengths for strong performance across diverse simulations. A qualitative example of a vital sign simulation of E2P-T+S is shown in Appendix C. Within the subset of shared tasks, our model achieves competitive or superior results compared to ETHOS, and further enables a wide range of tasks beyond its scope due to comprehensive data integration. These results highlight our strength in temporal modeling and simulation of patient pathways.

## 5.3 FINE-TUNING FOR OUTCOME PREDICTION

While our primary focus is on trajectory simulation, we also investigate the adaptability of our model as a foundation for outcome prediction tasks through targeted fine-tuning. We report results for Specialized Pathway Fine-Tuning (E2P-FT-P) and Outcome-Oriented Fine-Tuning (E2P-FT-O), and the specialized outcome prediction models MEME (Lee et al., 2024) and REMed (Kim et al., 2024) in Table 3. Both fine-tuning strategies demonstrate the model's ability to effectively specialize for specific outcomes, with the outcome fine-tuning outperforming pathway fine-tuning at the cost of of lacking intermediate outcomes. Compared to the baselines, EHR2Path-FT-O outperforms in three of four tasks. These findings underscore the versatility of our approach beyond trajectory prediction, highlighting its potential as a robust foundation for a wide range of EHR-based prediction tasks.

We further analyze model behavior across time and tasks. Appendix B reports performance as a function of prediction horizon, showing an expected degradation with longer rollouts, more pronounced for event prediction than for numerical values, yet remaining moderate up to 24 hours. Qualitatively, most errors arise from compounding deviations at longer horizons, missed very rare events, and underestimation of abrupt changes in otherwise stable numerical trajectories, while discharge timing errors tend to be delayed rather than prematurely predicted. For ICD prediction, which is a multi-label task, a 1-vs-all confusion analysis indicates that the main confusions concentrate in a few categories (e.g., "nervous", "digestive", "ill-defined" and "mental"), whereas most other categories do not exhibit systematically elevated false positive or false negative rates.

## 5.4 ADDITIONAL ABLATION RESULTS

To assess the impact of key components, we conduct ablation studies on the *Length-of-Stay (LOS) Indicator* and the *Masked Summarization Bottleneck* size. We assess the LOS Indicator's impact on tasks requiring convergence to final states, *ED Admission Prediction* and *Hospital Discharge Diagnosis*. Failure to converge within realistic, dataset-derived step limits is counted as an incorrect prediction. As shown in Table 4, removing the LOS Indicator significantly hinders convergence to a final state, reducing predictive accuracy. Further, we examine performance across various Masked Summarization Bottleneck sizes, trained for a limited number of steps. While longer contexts add detail, their quadratic cost limits scalability. Table 5 shows that, while larger bottleneck sizes slightly reduce validation loss, the most substantial improvement occurs from 4 to 8 tokens. Based on this, we select a bottleneck size of 8, balancing performance and efficiency.

Table 4: Effect of LOS Indicator.

|  |  | no LOS | with LOS |
|---|---|---|---|
| *Hospital Discharge Diagnosis* | F1 | 0.32 | **0.50** |
|  | converged | 51% | 100% |
| *ICU LOS* | Acc. | 0.65 | **0.69** |
|  | converged | 29.6% | 68.5% |

Table 5: Effect of Bottleneck Size.

| Size | Val Loss | Avg./Max. Tokens |
|---|---|---|
| 1 | 0.27 | 11 / 91 |
| 4 | 0.27 | 42 / 364 |
| 8 | 0.21 | 86 / 728 |
| 16 | 0.20 | 172 / 1456 |
| 32 | 0.18 | 344 / 2912 |
| 64 | 0.18 | 688 / 5842 |

## 5.5 LIMITATIONS AND FUTURE WORK

While the EHR2Path models establish a strong foundation for patient trajectory modeling, the focus on a single healthcare system may limit the generalizability of results to other healthcare settings, with differing demographics and site-specific treatment strategies, which could be addressed by collecting and using more comprehensive EHR data from a diverse set of countries and hospitals. Further, expanding the approach to include lifetime health data for long-term trajectory modeling is an exciting future direction.

## 6 CONCLUSION

In this work, we introduced a novel approach to patient pathway modeling, leveraging a structured text-based representation of heterogeneous EHR data and a Masked Summarization Bottleneck to handle extended data categories and temporal contexts efficiently. By integrating diverse data modalities, including numerical, categorical, and free-text information, our model enables accurate next-step predictions and robust simulation of patient trajectories over extended time horizons. Our experiments demonstrate that this approach provides a scalable framework for holistic patient modeling. By evaluating both next-step and long-term simulations, we highlight the potential of our method to improve predictive healthcare and personalized care. This work establishes a foundation for further research into comprehensive, task-independent EHR modeling, paving the way for future advancements in personalized and predictive medicine.

## REPRODUCIBILITY STATEMENT

We have taken several measures to make our study easy to replicate. All raw EHR data come from the public MIMIC-IV v2.2 release, and we describe the used tables and preprocessing steps in Section 4.1 and Appendix A of the paper. Data representation details are included in Section 3.1. Model and Training details are included in Section 3.2.1 and Appendix E. Detailed task definitions are detailed in Appendix G. Further, we will publish our complete code containing preprocessing scripts, model code, training pipelines, and evaluation scripts, as stated in the abstract.

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

APPENDIX

## A    DETAILS ON MIMIC-IV DATA

The MIMIC-IV Johnson et al. (2023b) dataset can be obtained via PhysioNet Goldberger et al. (2000); Johnson et al. (2023a) after performing the necessary credential process and CITI Data or Specimens Only Research training under the PhysioNet Credentialed Health Data License 1.5.0. We use version 2.2. of the dataset provided at `https://www.physionet.org/content/mimiciv/2.2/`, together with the corresponding versions of the MIMIC-IV-Note `https://www.physionet.org/content/mimic-iv-note/2.2/` and the MIMIC-IV-ED dataset `https://www.physionet.org/content/mimic-iv-ed/2.2/`) (Johnson et al. (2021)). In Table 6, we list all tables from the MIMIC-IV (Johnson et al. (2023b)) dataset we include into our patient representation as well as the data categories included in the ICU chartevents table:

Table 6: Used tables and data categories from the MIMIC-IV dataset.

| TABLE NAME | DESCRIPTION |
|---|---|
| **EMERGENCY DEPARTMENT** | |
| EDSTAYS | Information about patient admissions to the ED, including in and out times, as well as patient demographics. |
| MEDRECON | Current medications a patient is taking at admission. |
| TRIAGE | Information about the initial triage at arrival in the ED, such as admission vital signs and the chief complaint. |
| PYXIS | Dispensed medications during the ED stay. |
| VITALSIGN | Routine vital signs taken every 1-4 hours in the ED. |
| DIAGNOSIS | All billed diagnoses for a patient. |
| **HOSPITAL** | |
| OMR | Various measurements recorded outside the hospital. |
| ADMISSIONS | Information about hospital patient admissions, including partial demographics and admission and discharge times. |
| PATIENTS | Patients' age, gender and time of death. |
| SERVICES | The hospital service which cared for the patient in the hospital stay. |
| TRANSFERS | Information about transfers between different hospital unit. Can be different from the current service provider / care taker. |
| PRESCRIPTIONS | Prescribed medications during the hospital stay. |
| PROCEDURES | Performed procedures during the hospital stay. |
| LABEVENTS | Various laboratory measurements and test results. |
| MICROBIOLOGYEVENTS | Results of microbiology cultures. |
| DIAGNOSIS | Billed ICD-9/ICD-10 diagnoses for hospitalizations. |
| **ICU** | |
| ICUSTAYS | Information about ICU admissions, such as in and out time. |
| INPUTEVENTS | Information about continuous infusions or intermittent administrations in the ICU. |
| OUTPUTEVENTS | Information regarding patient outputs including urine, drainage, etc. |
| PROCEDUREEVENTS | Performed procedures during the ICU stay, such as ventilation or imaging exams. |
| CHARTEVENTS | Any charted items during the ICU Stay. We separate this data further into 20 categories: skin incisions, routine vital signs, skin impairment, cardiovascular, gi gu, toxicology, iabp, hemodynamics, respiratory, md progress note, adm history fhpa, dialysis, pain sedation, cardiovascular (pulses), pulmonary, cardiovascular (pacer data), nicom, alarms, skin assessment, neurological |

## B  PERFORMANCE ANALYSIS OVER FORECASTING HORIZONS

We report the results on the main development tasks across different forecasting horizons. We gradually increase the horizon in four-hour steps and analyze the effect on both event F1 and numerical accuracy. As expected, performance decreases with longer horizons, with a more pronounced drop for forecasting event occurrence (event F1) than for numerical values. Nevertheless, the degradation is moderate, and performance remains at a comparable level up to the full 24-hour forecasting horizon.

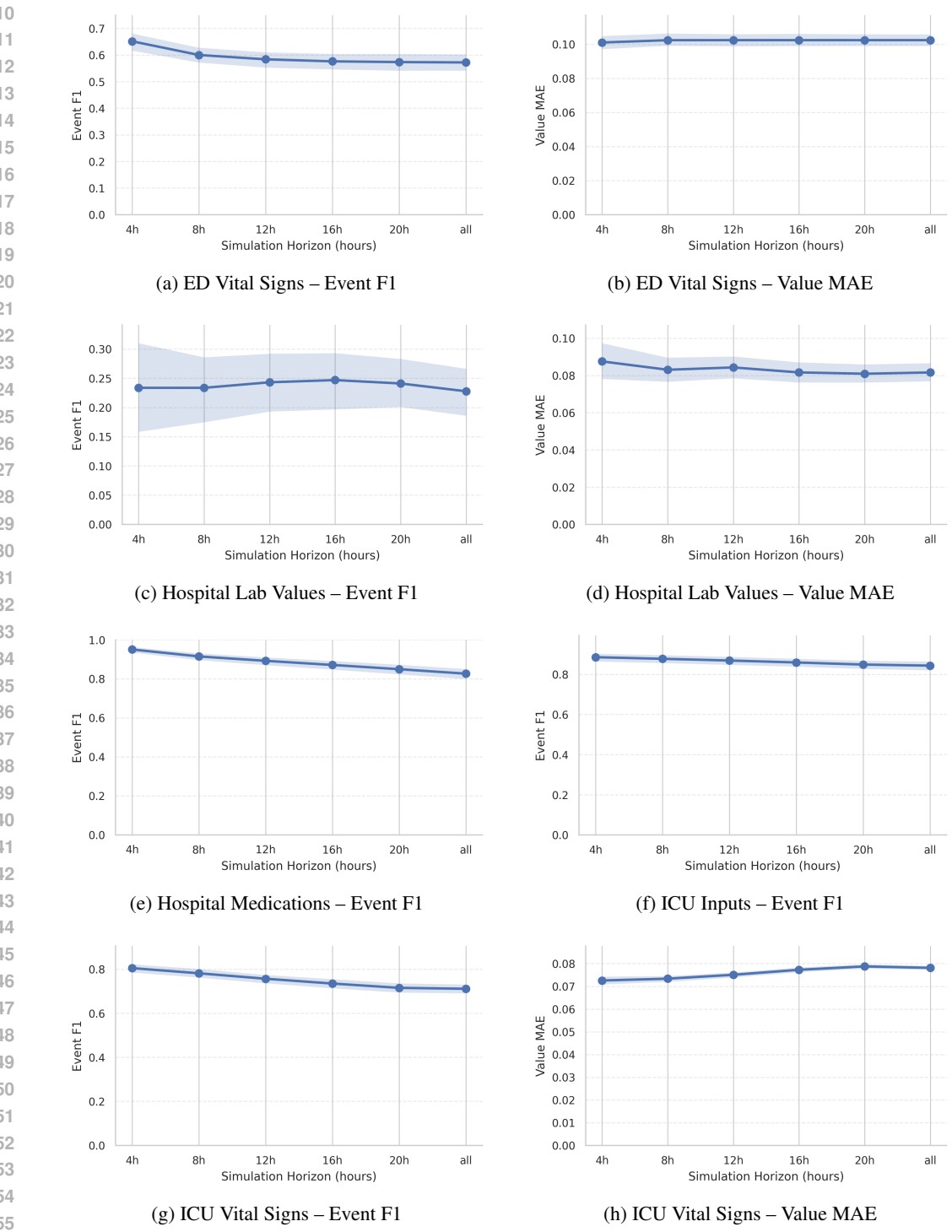

Figure 4: Development of event detection (F1) and value prediction (MAE) across simulation horizons for ED, hospital, and ICU development tasks. The shaded areas indicate 95% confidence intervals.

## C    QUALITATIVE EXAMPLE OF VITAL SIGN SIMULATION

Figure 5 shows a qualitative example of the predicted and real development of different vital signs of a patient, predicted by the combined model EHR2Path-T+S.

## D EXAMPLE OF OUR STRUCTURED TEXT REPRESENTATION

In Figure 6 and Figure 7, we show examples for the textual input and output representation for one time-point of a single patient.

## E IMPLEMENTATION DETAILS

Our model is implemented in PyTorch and based on the `Qwen-0.5b` model Yang et al. (2024), a compact LLM balancing capacity and efficiency, provided by unsloth Daniel Han & team (2023) under the apache-2.0 license (`https://huggingface.co/unsloth/Qwen2-0.5B-Instruct-bnb-4bit`). We train using a next-token objective with LoRA Hu et al. (2022) on structured text outputs (Section 3.1), optimized with AdamW ($1 \times 10^{-4}$ learning rate). At inference, we use the default Qwen decoding parameters ($temperature = 0.7, top_k = 20, top_p = 0.8$). Training runs on a single NVIDIA A40 GPU (48 GB) with a batch size of 8 for one epoch, covering one million time points. The summary-based model trains in 1.5 days, while the 24h text-based model and the combined model require approximately 4.0 days. The summarization model, trained separately on one million section samples, also completes in 4.0 days. The text-based model has a 4000-token limit. In the summary-based model, each section of max. 5000 tokens is compressed into 8 summary embeddings, achieving up to 625x compression. For the summary+text model, we leverage the summary-based model as starting point and fine-tune it using curriculum learning to leverage additional text inputs. During fine-tuning we randomly drop the text or summary inputs in 1/3rd of the training samples each. This incentivizes the model to leverage both input types. At inference time our simulation takes on average 3.5 seconds per simulated hour of data on a single GPU. For the LOS Indicator ablation, we set the convergence limits as follows giving a 24 hour buffer for convergence: for ED Admission, we cap simulation at 65 steps (24 hours beyond the longest ED stay in our test set), while Discharge Diagnosis simulations, which always occurs within one time step, are constrained to 24 steps. For all baselines, we use the original training and inference parameters. For all models, results are reported from a single run with bootstrapped 95% confidence intervals.

## F DETAILS ON WEIGHTED SAMPLING

In an EHR not all patient time points are equally informative. Many represent stable, uneventful progression, while a small subset contains rare but clinically significant changes. Uniform sampling leads to inefficient training and limits the model's ability to learn from these rare but important patterns. To address this, we apply weighting at the time point level, assigning a weight to each candidate timepoint (i.e., training sample) based on a relevance score, computed from the rarity of the events included in the output for that timepoint. The overall timepoint weight is formed as an average of the rarity of all events within that timepoint. We then sample training examples according to these weights, effectively enriching the training distribution with more informative and rare examples.

## G DETAILED TASK DESCRIPTIONS

**ED Tasks:**
*Vital Sign Development*: Predicts the progression of vital signs over 24 hours or until ED discharge, using data up to a random time point in the ED stay.
*Admission Prediction*: Determines whether an ED patient will be hospitalized or discharged home, using all static and dynamic data available at ED admission, simulating until the ED stay ends.
*Discharge Diagnosis*: Forecasts ICD categories at ED discharge (multi-label prediction over 18 categories as defined by Slee (1978)), using all ED data up to that point. The LOS indicator is set to zero for direct prediction of disposition and ICD categories.

**Hospital Tasks:**
*Medication Development*: Predicts administered medications over 24 hours or until hospital discharge, using data up to a random time point in the hospital ward.
*Lab Value Development*: Predicts performed lab tests and their results over 24 hours or until discharge, using data up to a random time point in the hospital ward.

*Discharge Diagnosis*: Forecasts ICD categories at hospital discharge (multi-label prediction over 18 categories as defined by Slee (1978)), using all patient data up to that point. The LOS indicator is set to zero for direct prediction.

**ICU Tasks:**

*Vital Sign Development*: Models ICU vital sign trajectories over 24 hours or until ICU discharge, using data up to a random time point in the ICU.

*Input Development*: Predicts administration of inputs (e.g., transfusions, medications) over 24 hours or until ICU discharge, using data up to a random time point in the ICU.

*Imminent Mortality*: Assesses whether a patient will die within 24 hours, using data up to a random time point in the ICU, with simulation continuing until discharge or death.

*Length-of-Stay (LOS)*: Predicts whether a patient will remain in the ICU for more than three days. Inputs include data up to a random time point, with simulation extending 72 hours or until discharge.

Table 7 summarizes input window, simulation horizon, and label definition for each task.

| Task | Task Type | Input Window | Gap Window | Output Window | Rolling/Static |
|---|---|---|---|---|---|
| ED Vital Signs | Development | up to random ED timepoint | 1h | 24h or until ED stay end | Rolling |
| ED Admission | Outcome | up to ED admission | – | ED stay end | Static |
| ED Discharge Diagnosis | Outcome | up to ED discharge | – | Direct prediction | Static |
| Hospital Medications | Development | up to random hospital timepoint | 1h | 24h or until hospital discharge | Rolling |
| Hospital Lab Values | Development | up to random hospital timepoint | 1h | 24h or until hospital discharge | Rolling |
| Hospital Discharge Diagnosis | Outcome | up to discharge | – | Direct prediction | Static |
| ICU Vital Signs | Development | up to random ICU timepoint | 1h | 24h or until ICU discharge | Rolling |
| ICU Inputs | Development | up to random ICU timepoint | 1h | 24h or until ICU discharge | Rolling |
| ICU Imminent Mortality | Outcome | up to random ICU timepoint | 1h | within 24h | Rolling |
| ICU Imminent Discharge | Outcome | up to random ICU timepoint | 1h | within 3 days | Rolling |

Table 7: Details of task definitions, including task type, input/ gap / output window and if the task is rolling or static.

# H    SOCIETAL IMPACT

Our holistic patient pathway model represents a promising advancement in healthcare technology by integrating diverse electronic health record (EHR) for longitudinal patient trajectory prediction. Such systems have the potential to enhance personalized and predictive patient care, leading to optimized treatment plans and better health outcomes. Ethically, this line of work necessitates strong protections for patient privacy and data security as it is based on sensitive patient information, as well as efforts to mitigate biases within the AI to ensure fair treatment for all patient groups. Societally, the model can support healthcare professionals by streamlining data analysis, making the vast amounts of patient data more manageable and helping in estimating effects of decisions in a patients treatment.

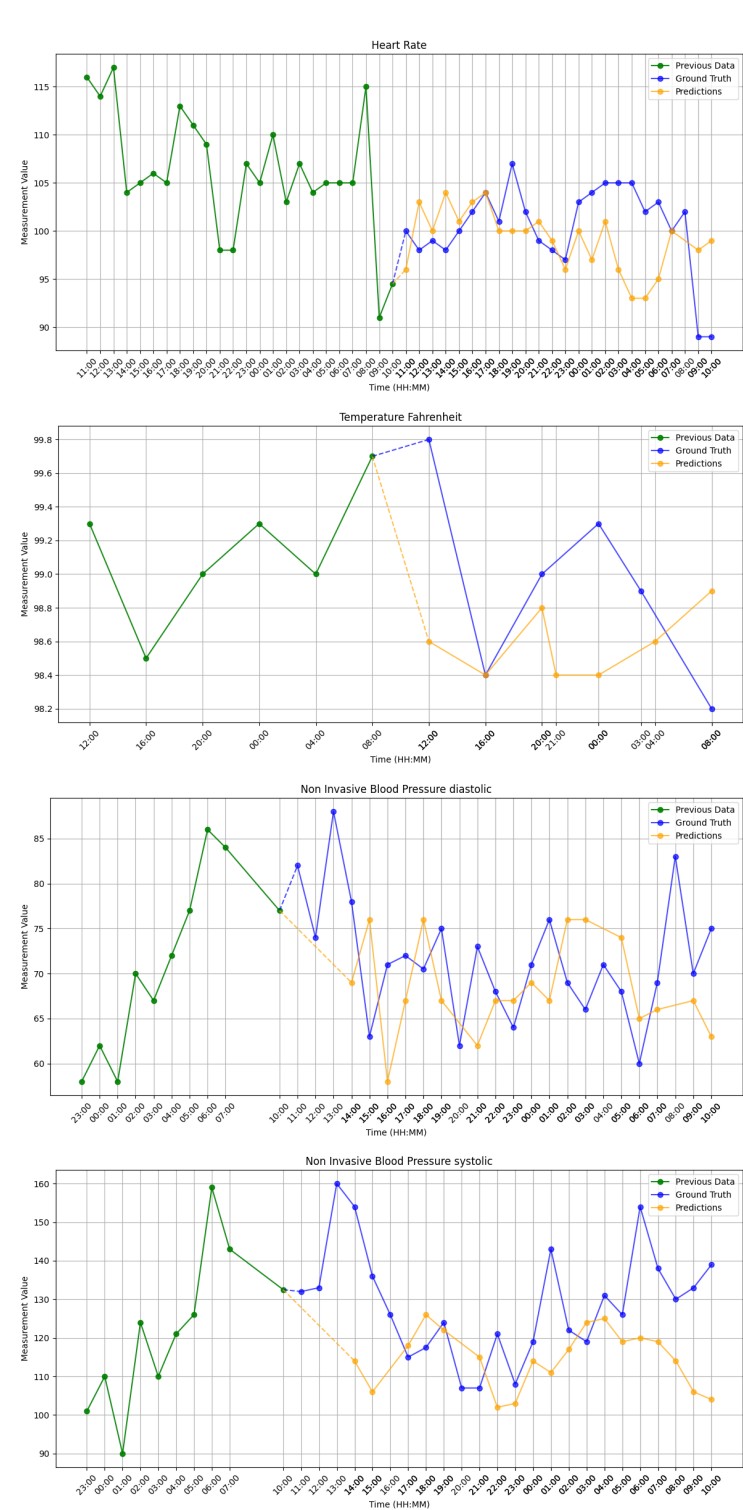

Figure 5: Qualitative Example of the ICU Vital Sign Simulation over 24 hours, showing ground-truth (blue), prediction (yellow) and 24h prior development (green) of Temperature, Heart Rate and diastolic and systolic blood pressure.

Figure 6: Example of structured input data from a patient record.

```
'Hospital Stay':
 'General': 'ew emer. patient, 57-year old male, insurance: Medicare,
    white, language:
   english'
 'Patient Location': '175-167: Emergency Department,167-135: Medical
    Intensive Care
   Unit (MICU),135-0: Coronary Care Unit (CCU)'
 'Care Taker': '169-3: Medical,3-0: Cardiac Medical'
 'Outpatient Measurements':
   'Height (Inches)': '174 days: 71'
 'Lab Results':
   'Lactate (mmol/L, normal range: 0.5-2.0)': '165: 7.8, 164: 7.6, [...]
      143/136: 2.2,
     133: 2.1, 122: 2.4, 119: 1.8'
 'Microbiology Growth Results':
   'GRAM STAIN - sputum': '145: no growth, 135: no growth'
 'Prescriptions':
   'sodium polystyrene sulfonate': '141-135'
   'dextrose': '97-79'
   'amiodarone hydrochloride': '97-74'
   [...]
 'Procedures':
   '7 days': 'Insertion of endotracheal tube; Continuous invasive
      mechanical ventilation
     for 96 consecutive hours or more; [...]'
 'Radiology Notes':
   '175': 'CHEST (PORTABLE AP): Limited study with new increased
      opacification of
     the left mediastinal contour and left heart border. Left-sided
        pleural effusion
     or focal consolidation cannot be excluded on this study. There is
        mild interstitial
     edema.'
'ICU Stay':
 'Stay 0':
   'Medication':
     'Heparin Sodium': '103-0'
   'Output':
     'Foley(ml)': '167-166: 0, 164: 10, 163-160: 0, 159: 40, 158: 15, 156:
        24, 153:
       80, 152: 20, 151: 30, [...], 5: 90, 3: 180, 1: 220'
   'Chart Events':
     'Cardiovascular':
       'RLE Color': '165-161: Mottled, 159-41: Normal, 36/33/29/21/17:
          Normal'
     'RoutineVitalSigns':
       'Arterial Blood Pressure systolic(mmHg)': '165: 71.5, 164: 118, 163:
          94, 162:
         89, 161: 92, [...], 11: 113, 10: 99, 9: 110, 7: 114.5,
         6: 106, 5: 104'
     'AdmHistory_FHPA':
       'Unable to assess teaching / learning needs': '130: 1'
     'Respiratory':
       'ETT Location': '166-108: Oral-R, 105-93: Oral-L, 89-77: Oral
          Center, 73/69/64/60/57:
         Oral Center'
     'Pulmonary':
       'Cough Effort': '167/117/113/109/41: Weak'
     'Skin-Assessment':
       'Braden Nutrition': '166: Probably Inadequate, 165-157: Adequate,
          153-145:
         Probably Inadequate, 141-137: Adequate, 134-80: Probably
            Inadequate, 72:
         Very Poor, 61-50: Probably Inadequate, 45/36/21/8/1: Probably
            Inadequate'
       [...]
```

(a) Example of an expected output for the next hour during the stay.

```
'Hospital Stay':
 'LOS': '116 hours'
 'Patient Location': 'Coronary Care Unit (CCU)'
 'Care Taker': 'CMED'
 'Prescriptions':
 - 'oxycodone hydrochloride and acetaminophen'
 - 'aspirin'
 - 'clotrimazole'
 [...]
'ICU Stay':
 'Stay 0':
  'LOS': '23 hours'
  'Medication':
  - 'Dextrose 5%'
  - 'Heparin Sodium'
  - 'Insulin - Regular'
  [...]
  'Procedures':
  - 'Multi Lumen'
  'Chart Events':
   'RoutineVitalSigns':
    'Heart Rate': '82.0'
    'Heart Rhythm': '1st AV (First degree AV Block) '
    'Non Invasive Blood Pressure diastolic': '78.0'
    [...]
   'Respiratory':
    'O2 saturation pulseoxymetry': '94.0'
    'Respiratory Rate': '17.0'
```

(b) Example of an expected output for the next hour at the end of the stay.

```
'Hospital Stay':
 'Disposition': 'DISCHARGED'
 'ICD categories': 'respiratory;pregnancy;congenital'
```

Figure 7: Examples of an expected output for the next hour during the stay (top) and at the end of the stay (bottom).

# I USE OF LARGE LANGUAGE MODELS

We used ChatGPT to improve the readability of the manuscript, specifically for grammar correction, shortening overly long sentences, and suggesting alternative phrasings. No text was inserted verbatim without manual review, and the tool did not generate any novel technical content, code, experimental results or figures.

