# OpenReview forum: "EHR2Path: Scalable Modeling of Longitudinal Health Trajectories with LLMs"
_ICLR.cc/2026/Conference — Submitted to ICLR 2026_

### Official Review · Reviewer_FtML · 2025-10-25

**Soundness:** 3
**Presentation:** 3
**Contribution:** 3
**Rating:** 6
**Confidence:** 4

**Summary:**

This paper presents EHR2Path, a transformer-based framework for modeling patient trajectories from EHR data using a structured text representation and a summarization mechanism that compresses long histories into compact tokens. The topic is timely and relevant, and the paper is well written and clear. However, several methodological and reporting limitations reduce the strength and interpretability of the results.

**Strengths:**

- Addresses an important and timely problem: scalable modeling of longitudinal EHR data using LLMs contributing to the growing shift from narrow outcome prediction toward holistic patient trajectory simulation.
- Conceptually interesting summarization mechanism that efficiently compresses long temporal contexts while maintaining continuity in modeling.
- Fairly novel unification of heterogeneous EHR modalities, structured, time series, and unstructured text into a textual representation, aligning with current trends in LLM-based clinical modeling.
- Generally well written and organized, with good motivation and consistent framing that make it accessible to both machine learning and clinical informatics audiences.

**Weaknesses:**

- Outputs only deterministic predictions without uncertainty estimates, limiting clinical usefulness. The absence of probabilities or confidence intervals prevents risk-aware decision support.
- Inference details are missing, decoding parameters such as temperature or top-k are never specified for either EHR2Path or ETHOS, hindering reproducibility and obscuring whether results depend on decoding strategy.
- Comparison with ETHOS is methodologically inconsistent: ETHOS uses stochastic Monte Carlo sampling, while EHR2Path is deterministic (?). Comparing them directly with deterministic metrics (F1, accuracy) is invalid without aligning inference methods.
- Inclusion of ETHOS results for both “restricted to data supported by ETHOS” and “full data” is contradictory, since ETHOS cannot process unstructured notes or ICU chart data. This raises concerns about what inputs were actually used.
- Test set details are incomplete: 5000 samples are mentioned, but class prevalences per task are omitted, making F1 and accuracy metrics difficult to interpret.
- Representation of diagnoses in the textual timelines is unclear, ICD categories are mentioned without specifying ICD-9 or ICD-10, undermining the claim of a unified representation. In addition, Appendix examples omit diagnosis tokens entirely, leaving this core modality undocumented.

**Questions:**

- What exact inference strategy was used, was temperature near zero or greedy decoding applied, or something else?
- How was ETHOS adapted, was Monte Carlo sampling not used or temperature modified, same question as above?
- For the full data ETHOS evaluation, what inputs were provided given unsupported modalities?
- What are the class prevalences for each downstream task?
- How are diagnosis categories represented textually, are ICD-9 codes converted to ICD-10 or standardized otherwise?
- Could you include an appendix example showing diagnosis representation in the timeline?
- Could you note in the appendix that the MIMIC-IV-ED extension was used and must be downloaded separately?
- Could you add a note in the Appendix that you used the MIMIC-IV-ED extension that has to be downloaded separately?

---

> ### Author Response · Authors · 2025-11-21
> **Rebuttal by Authors**
>
> We thank the reviewer for their feedback and for acknowledging that our work “addresses an important and timely problem”, “contributing to the growing shift from narrow outcome prediction toward holistic patient trajectory simulation”, our “conceptually interesting summarization mechanism” and the “fairly novel unification of heterogeneous EHR modalities”. In the following we address your open questions.
>
> **1. Uncertainty estimates**
>
> As proposed, we now added 95% confidence intervals to all quantitative results and metrics in Table 1-3. As we had to recompute some of the results, a few numerical values needed to be updated, but all qualitative conclusions and comparative outcomes remain the same.
>
> **2. Additional Implementation Details**
>
> We added the clarifications in the implementation details section in the revised paper (now Appendix E) and the task descriptions:
>
> *Decoding Parameters:* we use the default Qwen parameters for decoding (temperature=0.7, top_k=20, top_p=0.8). We added this to the implementation details section (now Appendix E).
>
> *Class prevalence in test set:* the 5000 samples are sampled without weighting from all timepoints, maintaining feature and class prevalence from the dataset. For most tasks, the F1 and accuracy metrics are macro averaged over a large set of features, where every feature has a different class prevalence, however many of the features are rather imbalanced and contain a large set of classes. For binary outcome tasks evaluated with accuracy, we use a balanced test set for sample efficiency. Discharge diagnosis is evaluated over 18 ICD categories. We added missing details to section 4.4.
>
> *Diagnosis representation:* During preprocessing, we unify ICD codes by converting all ICD-10 codes to ICD-9. For model prediction targets, we bucket all these ICD-9 codes into 18 coarse clinical categories as described in [1,2]. We clarify these details in the task description and added an additional outcome prediction example in the appendix, showing how diagnosis prediction is represented.
>
> *Use of MIMIC-IV-ED:* We added a note in Appendix A that we use this additional data source and added the citation and physionet link. It must be downloaded separately, but the same credential process is valid.
>
> [1] Vergil N Slee. The international classification of diseases: ninth revision (icd-9), 1978.
>
> [2] Matthew McDermott, Bret Nestor, Evan Kim, Wancong Zhang, Anna Goldenberg, Peter Szolovits, and Marzyeh Ghassemi. A comprehensive ehr timeseries pre-training benchmark. In Proceedings of the Conference on Health, Inference, and Learning, pp. 257–278, 2021.
>
> **3. Clarifications on ETHOS**
>
> For ETHOS, we keep all decoding parameters consistent with the published code (temperature 1.0 with sampling over all tokens, i.e., top_k=None). Both EHR2Path and ETHOS are evaluated with a single generated trajectory per patient–time point under these described decoding settings. For each task, we compute metrics over the full evaluation set and report 95% confidence intervals via bootstrap resampling over patients. To assess sensitivity to sampling noise in ETHOS, we repeated each experiment three times and observed only minor variation. In the original ETHOS work, multiple simulated futures per patient are primarily used to approximate probabilities for clinical outcomes and calculate AUROC metrics. ETHOS still needs to predict meaningful timeline simulations, thus this does not preclude evaluating ETHOS as a patient simulator with deterministic metrics.
> Regarding the full-data setting, ETHOS always receives exactly the input modalities it supports in the original work; the only difference to the restricted setting is that evaluation is performed over all modalities predicted by EHR2Path, including those not modeled by ETHOS. As we state in the paper, this mainly highlights that many features are not supported by ETHOS, leading to low F1 scores. For this reason, we additionally report a restricted comparison in which both models are evaluated only on the subset of features supported by ETHOS, where we still outperform.
>
> Thank you again for your helpful review. We hope our reply clarifies your open questions and will be helpful for your final evaluation.

---

> > ### Author Response · Authors · 2025-11-28
> >
> > Thank you again for your constructive feedback, which helped further improve the paper. As the discussion period is coming to an end, we would like to ask whether our rebuttal has addressed your concerns, or if there is anything else we could clarify.

---

### Official Review · Reviewer_DCx1 · 2025-10-29

**Soundness:** 3
**Presentation:** 2
**Contribution:** 2
**Rating:** 4
**Confidence:** 3

**Summary:**

This paper introduces EHR2Path, a novel model designed to address the challenge of modeling complex, longitudinal patient health trajectories using Large Language Models. The core problem it tackles is the failure of existing models to holistically interpret vast, heterogeneous Electronic Health Record data, which often comes from different clinical units like the Emergency Department (ED), hospital wards, and Intensive Care Unit. EHR2Path's solution involves transforming this diverse data—including structured fields, lab values, and unstructured text—into a unified, structured text representation. The key innovation is a novel "Masked Summarization Bottleneck," a mechanism that efficiently compresses long-term patient histories into topic-specific summary tokens, making the model highly token-efficient while capturing extended temporal context. The authors demonstrate that EHR2Path outperforms baselines on the MIMIC-IV dataset, showing strong performance in both next time-step prediction and the simulation of detailed future patient trajectories, such as forecasting vital signs and lab results.

**Strengths:**

A primary strength of EHR2Path is its comprehensive and scalable approach to data integration. Unlike previous models that often focus on a narrow set of features or specific outcomes , EHR2Path is designed to holistically process a patient's entire hospital stay, integrating all available structured and unstructured data from the ED, hospital, and ICU. The model's most significant technical strength is its novel Masked Summarization Bottleneck. This mechanism effectively solves the long-context problem in EHRs by compressing extensive patient histories into a compact set of summary tokens, allowing the model to be "much more token-efficient" than text-only approaches while leveraging long-term dependencies. This efficiency and holistic design translate to strong empirical performance, as the model not only outperforms trajectory simulation baselines but also proves to be a versatile foundation, outperforming specialized outcome prediction models when fine-tuned for those specific tasks

**Weaknesses:**

I didn't quite grasp the main contribution of this paper.

Predicting patient disease progression trajectories is an extensively studied problem. Researchers have commonly used models like RNNs, Transformers, and even Neural ODEs to forecast the temporal trajectories of high-dimensional data. To my understanding, these existing methods have already achieved a relatively high level of performance. Furthermore, techniques such as the Gumbel-Softmax trick already enable the joint prediction of continuous and categorical variables.

As I see it, the paper's motivation is not very clear. I don't fully understand the rationale for using an LLM for this task. It appears that the objectives of this paper can be achieved by existing methods, which can also be extended to handle unstructured text by incorporating embedding models. A particular drawback is the model's suboptimal performance; it does not seem to have been trained to a level sufficient for practical application.

**Questions:**

I noted in the Appendix that the training was performed based on a 0.5B quantized model. I am curious about the decision to perform SFT on a 4-bit model. Does applying SFT directly to a quantized version of such an extremely small model introduce any specific problems?

---

> ### Author Response · Authors · 2025-11-21
> **Rebuttal by Authors**
>
> We thank the reviewer for their feedback and for highlighting the “comprehensive and scalable approach” of EHR2Path to “holistically process a patient's entire hospital stay”, the “novel Masked Summarization Bottleneck” as the “most significant technical strength”, and our “strong empirical performance”. We appreciate the opportunity to clarify the main contribution, the motivation for using an LLM, and our quantization choice.
>
> **1. Main contribution and motivation for using an LLM.**
>
> Our goal is to jointly interpret and forecast the full next EHR state over an entire hospital stay in a single model. Classical RNNs/Transformers/Neural ODEs typically operate on a relatively small, hand-engineered state vector with heavy preprocessing. The main reason in using an LLM based approach over methods such as RNNs, Transformers, or Neural ODEs, is **handling heterogeneity via language:** As the reviewer noted in the "Strengths" section, the core challenge we target is "holistically interpreting vast, heterogeneous EHR data." By using an LLM, we utilize natural language as a unified interface, allowing the model to process various feature names and categorical, numerical, and free-text data natively without complex, lossy tokenization pipelines. They can also naturally deal with sparse and noisy inputs and outputs , and leverage their pre-trained knowledge for semantic understanding. The introduced Masked Summarization Bottleneck then enables full context handling by enforcing a pathway-focused, fixed-size information bottleneck. This combination of holistic problem formulation, unified representation, and bottlenecked LLM architecture goes beyond the application of existing sequence models.
>
> **2. Clarification on Performance**
>
> We respectfully note that we do not claim the model is currently ready for clinical deployment; that would require extensive prospective and external clinical validation beyond the scope of this work. In the context of machine learning research, however, we show that EHR2Path consistently outperforms strong baselines while handling a broader, more heterogeneous prediction space, as also reflected in the reviewer’s own remark about “strong empirical performance”. We believe this establishes EHR2Path as a strong new approach for holistic EHR trajectory modeling. We will clarify in the paper that our claims concern methodological and empirical advances, not immediate clinical deployment.
>
> **3. Choice of base LLM**
>
> We chose a 0.5B-parameter 4-bit model mainly for computational reasons, as our long-context setup leads to very long training times with larger models. We use a QLoRA-style setup: the base weights are stored in 4-bit and dequantized on the fly for matrix multiplications, while the LoRA adapter weights and forward-pass activations are kept in higher precision (fp16/bf16) during fine-tuning. This follows common practice in efficient LLM adaptation. In our experiments we did not observe specific instabilities due to 4-bit quantization. We will clarify this setup in the appendix.
>
> Thank you again for your feedback. We hope our reply clarifies your open questions and will be helpful for your final evaluation.

---

> > ### Comment · Reviewer_DCx1 · 2025-11-25
> >
> > I have carefully read the author's rebuttal, but I have decided to maintain my score.

---

> > > ### Author Response · Authors · 2025-11-25
> > >
> > > We would kindly ask you to let us know, which of your concerns were not addressed by our rebuttal.

---

### Official Review · Reviewer_AHTi · 2025-10-31

**Soundness:** 3
**Presentation:** 3
**Contribution:** 3
**Rating:** 4
**Confidence:** 4

**Summary:**

The paper proposes EHR2Path, a model that represents electronic health record (EHR) data as structured text and predicts patient health trajectories over time. It integrates heterogeneous data sources such as vital signs, lab tests, and clinical notes within a unified language-based representation. A Masked Summarization Bottleneck compresses long-term patient history into compact embeddings, enabling efficient modeling of extended timelines. The model can simulate future states of patients and supports both trajectory forecasting and outcome prediction. EHR2Path is evaluated on the MIMIC-IV dataset, where it outperforms baseline methods like ETHOS in next-step prediction and long-term simulation, demonstrating improvements in forecasting tasks and generalization across different hospital settings.

**Strengths:**

Here are the strengths of the paper in my opinion:

1. The paper tackles an important problem: modeling longitudinal patient trajectories rather than isolated outcomes. This is very important step towards clinical AI as it resembles the standard approach clinicians take too.
2. The proposed Masked Summarization Bottleneck is a neat adaptation for handling long patient histories.
3. Evaluation is conducted on a publicly availble dataset (MIMIC-IV) with multiple baselines and clearly defined metrics.

**Weaknesses:**

While the work is technically sound and relevant, but the contributions and empirical analysis have limitations. The major weaknesses of the paper are as follows:

1. The novelty is incremental: the paper mainly combines existing concepts (EHR tokenization, transformer summarization, next-step prediction) without introducing a fundamentally new modeling idea.
2. The experiments are limited to one dataset, and there’s little analysis on generalization, interpretability, or clinical significance of the forecasts.
3. The discussion of results is brief; there is minimal exploration of failure cases, ablations, or design justification beyond empirical comparison.
4. Presentation is adequate but somewhat dense, and the framing could be clearer in how this work advances prior EHR modeling approaches.

**Questions:**

To me although the methodology looks nice and the paper is tackling the modeling from an important perspective, there are many key downstream experimental points that are missing to make the work strong:

1. You state that the “Masked Summarization Bottleneck” compresses longitudinal context and forces outputs to attend only to summary tokens, with information-bottleneck framing. How does this differ in practice from known summarization or prefix-token schemes that restrict attention using masks or adapter tokens? Can you provide an ablation that replaces the custom mask with a simpler alternative?
2. The method appends m summary tokens per section and uses a mask to form a bottleneck. What is the performance vs. m trade-off and the compute cost vs. m trade-off? A plot of accuracy and wall-clock vs. m would help.
3. All experiments are on MIMIC-IV. A major weakness is that the authors do not provide stronger evidence for generalization. Can you add even a small held-out institution or a temporal split to demonstrate robustness? If that is not possible, can you simulate distribution shift in MIMIC-IV by patient subgroups or time periods and report stability?
4. Another suggestion is to use other publicly available EHR datasets to see how well the model generalizes there. To provide some examples, please check out NeurIPS D&B of the past 2 years. Specifically: https://som-shahlab.github.io/ehrshot-website/ can be a good candidate for the extra study.
5. Tables 1-3 present point estimates only. Please add confidence intervals or standard errors across seeds for next-step metrics and across simulation runs for longitudinal tasks.
6. How is LOS computed at training time without leaking future discharge timing into features that predict discharge?
7. How sensitive are simulation outcomes to errors compounding over long horizons? A breakdown of performance by simulation length would help.
8. You avoid heavy preprocessing, retain noise, and use natural language feature names and values. Could you show an explicit example that compares your textualization to code-based tokenization for the same patient slice, and then quantify the effect on accuracy and token counts?
9. Please include representative success and failure simulations for ICU vitals, inputs, and discharge timing, and discuss common error modes.
10. For diagnosis prediction tasks, show confusion matrices or top confusions for ICD categories.
11. Given that you are dealing with patient trajectories and you retain missing and incorrectly populated fields to embrace real-world noise, can you please quantify robustness: create stress tests that increase missingness or introduce synthetic noise and report degradation curves for key tasks.

---

> ### Author Response · Authors · 2025-11-21
> **Rebuttal by Authors**
>
> We thank the Reviewer for the detailed and constructive feedback, and for highlighting the importance of “modeling longitudinal patient trajectories rather than isolated outcomes”, the usefulness of the Masked Summarization Bottleneck, and the clarity of our experimental setup. We address your main points below:
>
> **1. Novelty and specifics of the masked summarization bottleneck**
>
> We respectfully disagree that our contributions are merely combinations of existing concepts. EHR2Path formulates **short-term patient pathway modeling as joint generative forecasting** of the full next EHR state, using a unified structured textual representation that, to our knowledge, for the first time covers the full MIMIC-IV hospital stay (22 tables and 20 ICU chart categories). This scale is enabled by our **Masked Summarization Bottleneck**, which enforces a hard, fixed-size information bottleneck via a custom attention mask: output tokens can only access the raw history through a small set of learned summary tokens that are trained end-to-end for forecasting rather than reconstruction. As a result, the model is forced to compress thousands of noisy, heterogeneous tokens into a dense latent representation, achieving up to 625× compression while improving forecasting performance. We are not aware of prior work that applies such a masked, forecasting-oriented information bottleneck to EHR trajectory modeling at this scale; if the reviewer has specific references in mind, we would be happy to cite and discuss them in the revised version.
>
> **2. Use of MIMIC-IV and generalization**
>
> We acknowledge reliance on a single healthcare system may constrain generalizability, as noted in our limitations section. However, using MIMIC-IV was a deliberate choice: it provides large-scale, granular trajectories across full hospital stays (ED, ward, ICU) with high-resolution vitals, labs, medications, inputs, notes, and various chart events, which matches our goal of joint forecasting of the full next EHR state. Other public datasets such as EHRShot are highly valuable and complementary, but they focus primarily on curated structured code features and cover a smaller set of feature sources. While EHR2Path could in principle be applied to such datasets by modifying the tasks and preprocessing, we believe, the scale and inherent real-world noisiness and sparsity of MIMIC-IV make it a more realistic benchmark for evaluating model performance under practical conditions.
>
> **3.  LOS computation without leakage**
>
> The LOS is computed as a simple countdown of remaining hours from the current timestep to the end of the stay. During training, the model must predict the next LOS value autoregressively, re-estimating remaining time. To increase robustness, and to allow the model to work without a ground truth LOS Token in inference time, we perform two augmentations. In half of the samples, we drop the LOS token entirely from the input, and in the other half, we inject noise (+-20%) into the LOS token, encouraging the model to re-estimate the remaining LOS at each time step. In inference/test, no ground-truth LOS is available, so the model starts without it and relies solely on its own predicted LOS values from prior autoregressive steps. **Crucially, at inference time, we never include the groundtruth LOS token in the input, so there is no data leakage.** Only the predicted LOS token of prior steps is included, starting with an input without LOS token in the first step. We clarified this in the method section about the Length-of-Stay Indicator.
>
> **4. Ablation on number of mask tokens**
>
> We already provide an ablation on the bottleneck size *m* in Table 5. As full training of summary and pathway models and evaluation for all *m* would be prohibitively expensive, we train only summary models with different *m* for a fixed, reduced number of steps and compare their validation loss. Increasing *m* improves validation loss but with diminishing returns, while the number of processed tokens grows linearly, increasing compute cost. The largest gain comes at *m=8*, after the loss plateaus despite higher cost, so we choose *m=8* as a good performance–efficiency trade-off.

---

> > ### Author Response · Authors · 2025-11-21
> > **Rebuttal by Authors (Part 2)**
> >
> > **5. Additional analysis and experiments**
> >
> > Thank you for the valuable suggestions on further results. We conducted the following additional analysis and experiments based on your proposals:
> >
> > *a) Confidence intervals*
> >
> > We now added 95% confidence intervals to all quantitative results and metrics in Table 1-3. As we had to recompute some of the results, a few numerical values needed to be updated, though all qualitative conclusions and comparative outcomes remain the same.
> >
> > *b) Performance by simulation length*
> >
> > We added an ablation on the change in performance with varying prediction horizons in the new Appendix B of the paper.  We report results on the main development tasks across different forecasting horizons by gradually increasing the horizon in four-hour steps and analyzing the effect on both event F1 and numerical accuracy. As expected, performance decreases with longer horizons, with a more pronounced drop for forecasting event occurrence (event F1) than for numerical values. Nevertheless, the degradation is moderate, and performance remains at a comparable level up to the full 24-hour forecasting horizon.
> >
> > *c) Discussion of failure cases*
> >
> > We added a discussion of observed error patterns in the revised version. Firstly, performance degrades with increasing simulation horizon, especially for the prediction of the next events, where deviations grow as we roll out further. Secondly, very rare events (e.g., very infrequent interventions or tests) are missed often and sometimes not predicted at all. Third, for numerical variables the dominant failure mode is extrapolating stable or slowly varying trends, leading to underestimation of sudden changes in some cases. For discharge timing, errors more often manifest as delayed prediction of death or discharge, rather than too early discharge or death.
> >
> > *d) Confusion analysis for ICD prediction tasks*
> >
> > We noticed it was not clear in the paper that ICD prediction is a multi-label task, therefore a direct confusion analysis between classes is not possible. Instead, we performed a 1-vs-all confusion analysis for both ED and Hosp ICD prediction. The results show that for hospital predictions, the categories the model confuses most often are “nervous”, “digestive” and “ill-defined”, with main errors being FPs and “mental” with the main errors being FNs. The rest of the categories have neither a comparatively large number of FPs or FNs. For the ED, mainly “ill-defined” has a larger number of FPs than the rest. We made it clear in the revised manuscript, that this is a multi-label task and if desired, we can include the 1-vs-all confusion matrices in the appendix.
> >
> > *e) Textualized vs coded representation*
> >
> > In our representation, each feature is named once and followed by its time–value pairs, allowing us to include non-coded chart events (e.g., reports, nursing assessments, wound dressing status), admissions history and exact numerical values and timestamps, for example:
> >
> > ~~~
> > LabResults: "Lactate (mmol/L, normal range: 0.5–2.0)": "165: 7.8, 164: 7.6, 143: 2.2"
> > RoutineVitalSigns: "Arterial Blood Pressure systolic (mmHg)": "166: 71.5, 164: 118, 163: 94, 162: 89, 140: 113".
> > AdmHistory_FHPA: "Unable to assess teaching / learning needs": "130: 1"
> > ~~~
> >
> > In contrast, coded event-based schemes usually enumerate all events in strict temporal order and repeatedly emit feature codes and discretized value buckets, e.g.
> >
> > ~~~
> > <Vital_ABP><Value_ABP:percentile_N> <Time:1h> <LAB_Lactate><Value_Lactate:percentile_N> <Time:1h> <LAB_Lactate><Value_Lactate:percentile_N> <Vital_ABP><Value_ABP:percentile_N> <Time:1h> <Vital_ABP><Value_ABP:percentile_N> <Time:>12h><LAB_Lactate><Value_Lactate:percentile_N><Time:1h><Time:1h><Vital_ABP><Value_ABP:percentile_N>
> > ~~~
> >
> > and cannot represent unstructured information such as the admissions history (AdmHistory_FHPA in the textualized sample). Tables 1 and 2 provide an indirect quantitative comparison: EHR2Path, using this textualization, consistently outperforms ETHOS on comparable pathway prediction tasks while modeling a richer set of features. If the reviewer finds this example helpful, we can include it in the appendix.
> >
> > Thank you again for your detailed review. We hope these clarifications address your questions and will be helpful for your final evaluation.

---

> > > ### Author Response · Authors · 2025-11-28
> > >
> > > Thank you again for your constructive feedback, which helped further improve the paper. As the discussion period is coming to an end, we would like to ask whether our rebuttal has addressed your concerns, or if there is anything else we could clarify.

---

### Official Review · Reviewer_Rnnx · 2025-11-01

**Soundness:** 3
**Presentation:** 3
**Contribution:** 2
**Rating:** 4
**Confidence:** 4

**Summary:**

This paper introduces EHR2Path, a scalable model using LLMs to predict longitudinal patient health trajectories. The method transforms diverse and heterogeneous EHR data, such as vitals, labs, and notes, into a unified and structured text representation. To efficiently handle long-term patient histories, the authors propose the ``Masked Summarization Bottleneck'', which compresses extensive temporal data into compact summary tokens. Based on the evaluation over MIMIC-IV dataset, EHR2Path demonstrates good performance that outperforms baselines in both next time-step prediction and the simulation of full patient trajectories.

**Strengths:**

1. This paper studies the problem of trajectory prediction with LLMs, which is an interesting topic in healthcare.
2. The paper is generally written with good clarity, and thus it is easy to follow.
3. The experimental results demonstrate the effectiveness of proposed method compared to the baselines.

**Weaknesses:**

1. One of the major concerns is the limited technical contribution. The proposed Masked Summarization Bottleneck and Length-of-Stay Indicator appear to be relatively straightforward techniques. These components function more as engineering solutions rather than novel methodological contributions, which raises questions about the paper's technical novelty.

2. Another major concern is the insufficient model evaluation. Specifically,

- a) The experimental comparisons for the Next-Timestep Prediction task are insufficient. Given that there are many existing models are capable of performing this task, a more comprehensive comparison is necessary. If the challenge lies in the mixed prediction types, the authors should consider conducting type-specific evaluations. For instance, isolating event prediction alone would be more feasible and would provide fine-grained insights into the model's performance across different prediction modalities.

- b) The evaluation methodology for trajectory prediction raises concerns as well. For time-sensitive outcomes such as mortality prediction, simple binary classification fails to capture prediction accuracy adequately, particularly since death occurs only once per patient, while the timing of predictions is critical in such scenarios but not considered. The experimental details in this section are unclear, and the evaluation approach requires substantial refinement.

3. One minor concern is the paper does not specify how many runs were conducted to obtain the reported experimental results. This is particularly critical for methods with LLM inference, as it is essential to assess the consistency or even hallucination concern of the model's predictions across multiple runs.

**Questions:**

1. Please refer to the above weakness for details.

---

> ### Author Response · Authors · 2025-11-21
> **Rebuttal by Authors**
>
> We thank the reviewer for their feedback and for recognizing the “good clarity” of the paper, the relevance of “trajectory prediction with LLMs” as “an interesting topic in healthcare” and the “experimental results [that] demonstrate the effectiveness of [the] proposed method compared to the baselines”. We address your open questions below:
>
> **1. Technical contribution**
>
> We respectfully disagree that our contributions are merely straightforward engineering. EHR2Path **formulates short-term patient pathway modeling as joint generative forecasting of the full next EHR state**, modeled as a unified structured textual representation for the first time covering the full MIMIC-IV records, including various information on admission, medications, procedures, reports, lab results and chart events. This scale is enabled by our **Masked Summarization Bottleneck**, which enforces a hard, fixed-size information bottleneck via a custom attention mask, trained end-to-end for forecasting rather than reconstruction. It addresses the fundamental challenge of scale and heterogeneity in EHR modeling by forcing the model to compress thousands of tokens of noisy history into a dense latent representation, achieving up to 625× compression. Furthermore, the **LOS token** with dropout and noise provides termination-aware conditioning for long-horizon rollouts: without it, convergence to realistic discharge/death trajectories collapses (100% → 51% hospital, 68.5% → 29.6% ICU; Table 4).
>
> **2. Evaluation**
>
> a) Table 1 already provides a type-specific breakdown, with separate metrics for event prediction (F1) and event+value correctness for numerical and categorical features, rather than a single mixed score. Our next-timestep task is defined as **jointly predicting the full next EHR state** over all static and temporal features. Most prior works are outcome-focused or focus on a small set of time-series and thus cannot generate complex, heterogeneous future values. Therefore, ETHOS is our primary baseline because it is one of the few models capable of this specific generative task. Table 2 focuses on task-specific evaluation and provides further comparisons to MEME and ReMed, which are specialized on specific tasks.
> b) For longitudinal simulation, development tasks (vitals, labs, medications, inputs) explicitly combine timing and values by computing event F1 and value errors over a 24h horizon (or until discharge) with a ±1h tolerance, while imminent mortality and discharge are evaluated as “within 24h / 3 days” labels, consistent with prior work. We will clarify these design choices in the revised version.
>
> **3. Details on number of runs**
>
> Finally, due to the cost of training and evaluation, we report a single training run per model. To still convey uncertainty, we now added 95% bootstrap confidence intervals to all our results. We added this information in the Implementation Details section (now Appendix E).
>
> Thank you again for your valuable review. We hope these clarifications address your questions and will be helpful for your final evaluation.

---

> > ### Author Response · Authors · 2025-11-28
> >
> > Thank you again for your constructive feedback, which helped further improve the paper. As the discussion period is coming to an end, we would like to ask whether our rebuttal has addressed your concerns, or if there is anything else we could clarify.

---

### Author Response · Authors · 2025-12-03
**Summary of Rebuttal Process**

Dear Area Chair,

We thank the reviewers for their time and valuable feedback. Given the special situation, and as three of our reviewers have not yet had reacted to the rebuttal before the rebuttal responses were disabled, we want to take this opportunity to provide a short summary of the reviews and the resulting changes and additions to our work.

The key topics discussed in the reviews were:

* a) Overall contribution and motivation for using an LLM
* b) Clarifications on the evaluation setup and baselines
* c) Lack of confidence intervals / uncertainty estimates
* d) Behavior over longer prediction horizons
* e) Discussion of typical failure cases
* f) Confusion analysis for ICD prediction tasks
* g) Concrete example of textualized vs. coded representations
* h) Implementation and training details

In our rebuttal and revised manuscript, we responded as follows:

* a) We **clarified our main contribution** is a scalable LLM-based framework for joint next-state forecasting over full EHR trajectories, introducing a novel masked, forecasting-oriented summarization bottleneck and LOS token to enable long-horizon, heterogeneous predictions. Further, we **provided a discussion on the reason for using an LLM**, which allows handling heterogeneity and sparsity via language.

* b) We highlighted the **task-specific evaluations** (events vs. event+value) in Table 2, **explained why ETHOS is our main baseline** as one of the few works targeting pathway prediction, and motivated the **deliberate choice of MIMIC-IV** as a large, noisy, full-stay benchmark.

* c) We **added confidence intervals to all results in Tables 1–3** to provide uncertainty estimates for our reported metrics and assess the model’s consistency.

* d) We introduced an **additional evaluation of performance by simulation length** where we gradually increase the prediction horizon for all development tasks (see Appendix B), illustrating how performance evolves with rollout length.

* e) We **added a qualitative discussion of typical failure cases** (e.g., rare events, long-horizon drift) in the main manuscript.

* f) We **performed and reported one-vs-all multi-label confusion analysis** for both ED and Hospital ICD prediction to clarify error patterns across diagnosis categories.

* g) We provided a concrete **example contrasting our textualized representation with a code-based alternative** to clarify what information is captured and why we chose the textualized design.

* h) We **extended our implementation details**, adding the number of evaluation runs our choice of base LLM and QLoRA setup, decoding parameters, class prevalences and the ETHOS evaluation protocol. Further, we **clarified specifics** on our mask-size trade-off ablation, how LOS is used without test-time leakage, and the diagnosis representation (with an additional output example in Appendix D).

We believe these additional analyses and clarifications substantially strengthen the paper and directly address the main concerns raised in the initial reviews. We again thank the area chair and the reviewers for their efforts.

Best regards,

The authors

---

### Meta-Review · Area_Chair_9DWJ · 2025-12-26

**Summary:**

The paper proposes EHR2Path to model the longitudinal health trajectories with LLMs. For this paper, 4 reviewers have uploaded their comments. At the very first begin,  there are 3 negative scores and 1 positive score.  The summary of reviewers' concerns are as follows.
1) For the Reviewer Rnnx, the reviewer  shows the concerns about the limited technological contributions and model evaluation and gives a negative score;
2) For the Reviewer  AHTi, the reviewer shows the concerns about the incremental novelty, limited dataset used, brief discussion of results, lack of clear comparason with related works. And the reviewer gives a negative score;
3) For the Reviewer  DCx1, the reviewer shows the concerns about unclear motivations;
4) For the Reviewer FtML, the reviewer shows the concerns about limited clinical usefulness, inconsistent comparison with ETHOS, and lack of experimental details.

The authors have made efforts to address the concerns raised by reviewers. However, the paper still suffer from the limitations such as the limited technological contributions,  unclear clinical usages. Therefore, the paper is rejected.

**Reviewer Concerns:**

For Reviewer Rnnx,  W3 may be fully addressed as the author appended experimental results as required. W1 and W2 are still outstanding.
For Reviewer AHTi, the reviewer focuses on more detailed information such as key downstream experimental points and listed 11 questions.  However, the rebuttals are not point by point. It can not be inferred whether the specific questions are perfect addressed.
For Reviewer DCx1, the author has pointed out the key contributions but the major concern is that technologies such as Gumbel-Softmax trick already enable the joint prediction of continuous and categorical variables. This concern is not fully addressed. Addtionly, from the comments of the reviewers, it is likely the author does not convince the reviewer.
For Reviewer FtML, most of the concerns are addressed.

**Reviewer Scores:**

For Reviewer Rnnx and AHTi, as they point out that the concerns of novelty, it is unlikely to change their score to be positive. For Reviewer DCx1, the reviewer has shown that he will main the final score as negative. For Reviewer FtML,  as the confidence is not very strong, it is unlikely to change the score.

---

### Decision · Program_Chairs · 2026-01-26

Reject